# Variational Quantum Linear Solver enhanced Quantum Support Vector Machine

## Abstract

Quantum Support Vector Machines (QSVM) play a vital role in using quantum resources for supervised machine learning tasks, such as classification. However, current methods are strongly limited in terms of scalability on Noisy Intermediate Scale Quantum (NISQ) devices. In this work, we propose a novel approach called the Variational Quantum Linear Solver (VQLS) enhanced QSVM. This is built upon our idea of utilizing the variational quantum linear solver to solve system of linear equations of a Least Squares-SVM on a NISQ device. The implementation of our approach is evaluated by an extensive series of numerical experiments with the Iris dataset, which consists of three distinct iris plant species. Based on this, we explore the effectiveness of our algorithm by constructing a classifier capable of classification in a feature space ranging from one to seven dimensions. Furthermore, we exploit both classical and quantum computing for various subroutines of our algorithm, and effectively mitigate challenges associated with the implementation. These include significant improvement in the trainability of the variational ansatz and notable reductions in run-time for cost calculations. Based on the numerical experiments, our approach exhibits the capability of identifying a separating hyperplane in an 8-dimensional feature space. Moreover, it consistently demonstrated strong performance across various instances with the same dataset.

## 1 Introduction

Support vector machines (SVMs) are one of the most renowned and widely used machine learning algorithms due to its ability to handle high dimensional data. It was initially formulated as a quadratic programming problem (Vladimir & Vapnik, 1998). The primary task of an SVM is to construct a separating hyperplane that classifies data in the feature space. While SVMs are effective for many tasks, they might not be as scalable as some other methods, such as the least square formulation of SVM (LS-SVM), especially for large datasets (Chua, 2003). The LS-SVM is a reformulation of SVM as a linear programming problem which is equivalent to solving a system of linear equations (SLEs), making it computationally less complex (Suykens & Vandewalle, 1999).

Rebentrost et al. proposed a quantum version of LS-SVM, known as the QSVM (Rebentrost et al., 2014). This method successfully computes the inverse of the feature matrix by leveraging the principles of the HHL algorithm, coming from Harrow, Hassidim, and Lloyd (HHL) (Harrow et al., 2009). HHL is designed to efficiently solve SLEs and its computational complexity scales logarithmically with respect to the system size. However, the implementation of the HHL poses significant challenges when it comes to the efficient execution on the current Noisy Intermediate Scale Quantum (NISQ) devices. This is primarily due to the extensive demand of quantum resources. Additionally, QSVM (Rebentrost et al., 2014) requires that the training data is prepared as a coherent superposition and provided as an imput to the quantum hardware for computing the inverse of the kernel matrix, thus making it a plausible algorithm only when implemented on a fault tolerant, large scale quantum computer.

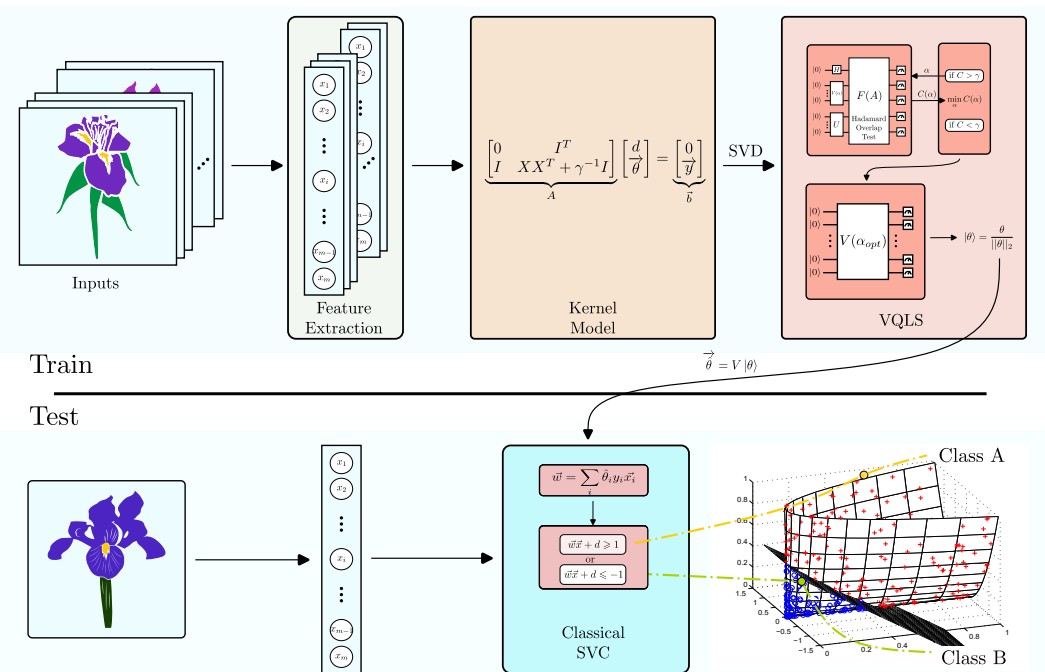

Figure 1: Pictorial representation of VQLS enhanced QSVM.

Thus, quantum classical hybrid algorithms were developed that are capable of efficiently solving a task partially on a NISQ computer. Variational hybrid quantum-classical algorithms (VHQCAs) are a class of such hybrid algorithms, where classical pre- and post-processing methods are combined with quantum subroutines. They have been used to solve a variety of physical problems varying from quantum chemistry to quantum machine learning (Kandala et al., 2017; Biamonte et al., 2017). The idea of VHQCAs is to use shallow quantum circuits for quantum subroutines combined with classical post processing or optimization techniques. In 2019, HavlÄśček et al. proposed a variational approach, where the authors estimated the kernel function on a quantum computer and subsequently optimized a classical SVM on the classical computer (Havlíček et al., 2019). However, this approach was assessed using a small toy dataset with just two features. Similar ideas were explored applying different classical optimization procedures based on gradient descent (Li et al., 2022), regularized Newton method (Zhang et al., 2022), and $\Gamma$ matrix expansion (Ezawa, 2022). QSVM has been realized experimentally on quantum hardware limited to two features (Li et al., 2015). Hence, this leaves an unexplored research area regarding the performance and practical scalability of QSVM when applied to larger-scale, real-world problems on NISQ hardware. This motivates our investigation presented henceforth.

We propose a novel approach within the realm of QSVM, the Variational Quantum Linear Solver enhanced QSVM (VQLS-enhanced QSVM). A pictorial representation of our algorithm is presented in Fig. 1. The idea of VQLS was proposed by Bravo-Prieto et al. (2019) as a hybrid quantum classical algorithm, designed to solve SLEs with a polylogarithmic scaling in problem size. VQLS has proven to be effectively scalable on NISQ devices for large problem sizes given a well conditioned, sparse matrix. However, to the best of our knowledge, the effectiveness of VQLS for solving SLEs with dense matrices derived from real-world datasets has not yet been investigated. To this end, we develop a classifier from VQLS-enhanced QSVM. We then evaluate the performance, by conducting an extensive series of numerical experiment using the Iris dataset (Fisher, 1988). These experiments were executed on IBM-Q simulators (Qiskit contributors, 2023) in the noise-free environment. We analyze the numerical results of our experiments and present strategies to mitigate the hurdles of utilizing VQLS-based QSVM for real-world applications. Based on the numerical analysis of our experiments, our VQLS-enhanced QSVM succeeded in identifying optimal

hyperplane parameters within an 8-dimensional feature space. This is further supported by the construction of support vector classifier (SVC) and the subsequent evaluation of its classification accuracy.

The paper is structured as follows: In Sec. 2, we briefly discuss the theory of SVMs and VQLS. In Sec. 3, we present our approach of combining the two ideas. Sec. 4 presents results and discussion, and finally conclusions are in Sec. 5.

## 2 Theoretical Preliminaries

### 2.1 Support Vector Machines

SVMs have long been a cornerstone of classical supervised machine learning, serving as a powerful tool for data classification in feature spaces (Vladimir & Vapnik, 1998). An SVM constructs a separating hyperplane that classifies data, illustrated in Fig. 1. An SVM is a quadratic programming problem and the least squares formulation in Suykens & Vandewalle (1999) proposes a method to obtain parameters via solving an SLE. In this section, we discuss briefly the least squares formulation of SVMs (LS-SVM). Given the tuple $\{y_k, \vec{x}_k\}_{k=1}^N$ as the training set of $N$ data points, the weights are given by $\vec{w}$ and the offset by $d$. The function $\varphi(\circ)$ is a map from the input vector space spanned by the training data to a higher dimensional space where classification is possible. Solving an SVM and finding the parameters for constructing the optimal hyperplane can be reformulated as an optimization problem with variables $\eta_k$ (Suykens & Vandewalle, 1999) in the following way:

$$\min_{\vec{w}, \eta_k} \mathscr{J}(\vec{w}, \eta_k) = \frac{1}{2}\vec{w}^T\vec{w} + c\sum_{k=1}^N \eta_k. \tag{1}$$

In which case, the separating hyperplane takes the form:

$$\begin{aligned} y_k[\vec{w}^T\varphi(\vec{x}_k) + d] &\geq 1 - \eta_k, \\ \eta_k &\geq 0, \qquad k = 1, \dots, N. \end{aligned} \tag{2}$$

In Suykens & Vandewalle (1999), the least squares version is introduced as

$$\min_{\vec{w}, d, \vec{e}} \mathscr{I}(\vec{w}, d, \vec{e}) = \frac{1}{2}\vec{w}^T\vec{w} + \gamma\sum_{k=1}^N e_k^2, \tag{3}$$

where $e_k$ corresponds to a set of slack variables which are inserted to get an equality sign instead of inequality in Eq. (2). Here, the separating hyperplane takes the form:

$$y_k[\vec{w}^T\varphi(\vec{x}_k) + d] = 1 - e_k, \qquad k = 1, \dots, N, \tag{4}$$

where $\gamma$ is a tunable hyperparameter. The optimization Lagrangian takes the form:

$$\mathscr{L}(\vec{w}, d, \vec{e}; \vec{\theta}) = \mathscr{I}(\vec{w}, d, \vec{e}) - \sum_{i=1}^N \theta_i(\vec{w}^T\varphi(\vec{x}_i) + d + e_i - y_i), \tag{5}$$

where $\vec{\theta}$ are the Lagrange multipliers. Optimality conditions correspond to the linear system defined in Suykens & Vandewalle (1999):

$$\begin{pmatrix} 0 & \vec{1}^T \\ \vec{1} & X^T X + \gamma^{-1}\mathbb{1} \end{pmatrix} \begin{pmatrix} d \\ \vec{\theta} \end{pmatrix} = \begin{pmatrix} 0 \\ \vec{y} \end{pmatrix}. \tag{6}$$

Here $\vec{1} = [1, \dots, 1]^T$ is a column vector of dimension $N$ and $\mathbb{1}$ is the $N$-dimensional identity matrix in the canonical basis. Once the hyperparameters such as $\gamma$ are fixed, the LS-SVM classifier is evaluated using the test data (Suykens & Vandewalle, 1999):

$$\hat{y}(\vec{x}) = \vec{w}^T\varphi(\vec{x}) + d = \sum_{i=1}^N \theta_i \varphi(\vec{x}_i)^T \varphi(\vec{x}) + d. \tag{7}$$

## 2.2 Variational Quantum Linear Solver

In this section, we summarize the essentials of the algorithm from Bravo-Prieto et al. (2019) solving SLEs by a variational approach. VQLS takes the following inputs: the state $|b\rangle$, the matrix representation of $A$ and the set of $\{\alpha_i\}$ as the initial set of parameters. For state initialization, there is a unitary operator that is able to efficiently execute $U|0\rangle = |b\rangle$ as a quantum circuit (Shende et al., 2005). And the given matrix $A$ is decomposed into a linear combination of unitary matrices,

$$A = \sum_{l=0}^{N} c_l A_l. \tag{8}$$

It is imperative that the condition number $\kappa$ of $A$ is finite, $\|A\| \leq 1$, and the unitary $A_l$ can be efficiently implemented by a quantum circuit. Generally, for qubit systems, $A_l$ can be further decomposed as a combination of Pauli strings $P_l$, where $P_l \in \{\mathbb{1}, X, Y, Z\}^{\otimes N}$.

### 2.2.1 Variational Ansatz

The solution state $|x\rangle$ is prepared by a quantum circuit as $|x\rangle = V(\alpha)|0\rangle$, where $V(\alpha)$ is a sequence of parameterized quantum gates for the chosen ansatz. The cost function $C(\alpha)$ is computed in the same circuit to estimate the overlap between $A|x\rangle$ and $|b\rangle$. A popular choice is the hardware efficient ansatz (Kandala et al., 2017) from the family of fixed layer ansatz. However, it is known to be hard to train (Wang et al., 2021; Cerezo et al., 2021). An overview of different ansatze is presented in Tilly et al. (2022).

### 2.2.2 Cost Functions

The global cost function is defined in Bravo-Prieto et al. (2019) as:

$$\begin{aligned} C_{global} &= \frac{1}{\langle\psi|\psi\rangle} \left[ \langle x| A^\dagger (\mathbb{1} - |b\rangle\langle b|) A |x\rangle \right] \\ &= 1 - \frac{|\langle b|\psi\rangle|^2}{\langle\psi|\psi\rangle}, \end{aligned} \tag{9}$$

where $|\psi\rangle = A|x\rangle$. Alternatively, a local cost function is proposed in Bravo-Prieto et al. (2019) which is resilient to Barren plateaus for large system sizes (Cerezo et al., 2021), as $n$ grows.

The cost functions are computed in the variational circuit by using the Hadamard test or the Hadamard overlap test. In terms of minimizing the number of controlled operations, the Hadamard overlap test is preferred at the expense of increasing the number of qubits in the quantum circuit. In this work, the values of quantities $\langle\psi|\psi\rangle$ and $|\langle b|\psi\rangle|^2$ are determined by using the Hadamard test. The first component is equivalent to computing (Bravo-Prieto et al., 2019):

$$\langle\psi|\psi\rangle = \sum_m \sum_n c_m^* c_n \langle 0| V(\alpha)^\dagger A_m^\dagger A_n V(\alpha) |0\rangle \tag{10}$$

Each term of the form $\langle 0| V(\alpha)^\dagger A_m^\dagger A_n V(\alpha) |0\rangle$ inside the sum of Eq. (10) is evaluated by controlled execution of $A_m^\dagger$ and $A_n$. The implementation of a quantum circuit for this term is presented in Fig. 3 in Appendix A.1.

Similarly, the computation of the second component is given by Bravo-Prieto et al. (2019),

$$|\langle b|\psi\rangle|^2 = \sum_m \sum_n c_m^* c_n \langle 0| U^\dagger A_n V(\alpha) |0\rangle \langle 0| V(\alpha)^\dagger A_m^\dagger U |0\rangle \tag{11}$$

Here, the implementation of two inner products $\langle 0| U^\dagger A_n V(\alpha) |0\rangle$ and $\langle 0| V(\alpha)^\dagger A_m^\dagger U |0\rangle$ inside the sum requires two more controlled operations of $U$, $V(\alpha)$ with $A_n$ and $A_m^\dagger$. Fig. 4 in Appendix A.1 illustrates the implementation of the term $\langle 0| V(\alpha)^\dagger A_m^\dagger U |0\rangle$.

### 2.2.3 Classical Optimization

To obtain an optimal set of parameters $\{\alpha_i^{opt}\}$, a classical optimizer is necessary. In Bravo-Prieto et al. (2019), gradient based optimization is used. In this work, we use gradient free optimizer, specifically, COBYLA (Powell, 1994). A comparison between different optimization methods for hybrid quantum classical variational algorithms is presented in Nannicini (2019) and Pellow-Jarman et al. (2021).

## 3 Algorithm

We take advantage of VQLS to solve Eq. (6), extract parameters $\{\alpha_i^{opt}\}$ to estimate the solution state $|x\rangle$, and construct a separating hyperplane. This hyperplane is further used for the classification of the samples in test dataset. A pictorial representation of our algorithm is presented in Fig. 1. Further specifications about the execution are discussed in this section. Additionally, the pseudo-code for our novel VQLS-enhanced QSVM algorithm is presented in Algorithm 1 and 2 in Appendix A.2.

### 3.1 Dataset

In this work, we use the Iris dataset (Fisher, 1988) to evaluate the effectiveness and feasibility of our algorithm. It contains 50 examples for each of the three distinct iris plant species, *Setosa*, *Virginica*, and *Versicolor*. Each sample is composed of four distinct attributes: sepal length, sepal width, petal length, and petal width, all quantified in centimeters. For our numerical experiments, two species, *Setosa* and *Virginica* have been selected. From these two species, a total of seven samples have been chosen randomly for the training dataset. Table 1 in Appendix A.3 presents a concise overview of a single instance of the utilized training dataset.

### 3.2 Data preprocessing and construction of kernel model

In order to prevent a particular feature from dominating the others due to its large magnitude, a data normalization technique known as linear scaling has been applied in our work, so that they all fall within the range of $[0, 1]$. It is worth highlighting that normalization significantly influences the trainability of variational ansatz, as detailed in Appendix A.4.

The normalization for a feature $x^j$ is given by:

$$x_{norm}^j = \frac{x^j(i) - x_{min}^j}{x_{max}^j - x_{min}^j},\tag{12}$$

where $i$ is the index of training samples.

The representation of the kernel matrix is formulated in Eq. (6). The dimension of the kernel matrix $K$ is $(N + 1) \times (N + 1)$, where $N$ is the number of samples in the training dataset. The presence of an additional row and column is a consequence of the non-zero offset $d$. In the context of the linear equation $A\vec{x} = \vec{b}$, the kernel matrix $K$ corresponds to the matrix $A$.

In designing hybrid quantum classical algorithms executed on current quantum hardware effectively, it is important to strategically distribute different parts of our algorithm on different computing platforms. For this reason, we use SVD prior to Pauli decomposition to reduce the number of controlled components of the kernel matrix, subsequently reducing the hard part of the calculation of the cost function. It is worthwhile to note that the Pauli decomposition in Bravo-Prieto et al. (2019) is executed on a classical computer as a one-time preprocessing step. Although there exist efficient methods to simulate such decomposition on a quantum computer (Heidari & Szpankowski, 2023; Montanaro & Osborne, 2019), the comparison of resource overhead has not been explored in the context of employing them for variational algorithms. To that end, we aim to enhance the performance of VQLS by introducing SVD. This step is crucial towards the trainability of the variational ansatz we

use and the reduction in the time for training as we will discuss in Sec. 4. Hence, we recast the problem as follows:

$$A \ket{x} = W \Sigma V^T \ket{x} = \ket{b}. \tag{13}$$

The above can be reformulated as :

$$A_{new} \ket{x_{new}} = \ket{b_{new}}, \tag{14}$$

where $A_{new} = \Sigma$, $\ket{b_{new}} = W^T \ket{b}$. In case of the termination of the algorithm, the estimated state is related to our solution by $V^T \ket{x_{new}} = \ket{x}$.

Hamiltonian decomposition is a pivotal factor when it comes to variational algorithms in determining plausible effectiveness. Hence, various methods offer efficient Hamiltonian decomposition (Berry et al., 2007; Childs et al., 2017), particulary when the Hamiltonian exhibits the sparsity. Extending the same framework to our kernel matrix, it is imperative to improve the sparsity by employing SVD for an efficient quantum subroutine.

### 3.3 Implementation of VQLS

Building upon the basic implementation of VQLS detailed in Qiskit contributors (2023), we extend its functionality to implement our VQLS-enhanced QSVM.

#### 3.3.1 Variational Ansatz

The Ansatz $V(\alpha)$ in the VQLS is realized by using a hardware-efficient ansatz designed for a three-qubit circuit, as introduced in Bravo-Prieto et al. (2019). The quantum circuit of this hardware-efficient ansatz, initialized with random parameters, is shown in Fig. 5 in Appendix A.1.

#### 3.3.2 Quantum circuit for computing the cost function

The state $\ket{x}$ is prepared with the ansatz by $V(\alpha) \ket{0}$. The value of the cost function indicates the overlap of $A \ket{x}$ with the solution state $\ket{b}$. A higher cost indicates a lower overlap between current and desired solution. Therefore, it is crucial to determine an optimal set of the parameters $\{\alpha_i^{opt}\}$ through an optimization method on a classical computer by minimizing the cost function from Eq. (9). Details of the code to compute the cost function are explained in the pseudocode presented in Algorithm 1.

#### 3.3.3 Construction and validation of SVC

The set of optimal parameters $\{\alpha^{opt}\}$ obtained through Algorithm 1 in Appendix A.2 is delivered to initialize the hardware-efficient ansatz, allowing us to estimate the vector $\vec{\theta}$ after measurement.

The measured probabilities of each basis state in the statevector indicate the weights of $\vec{x}_k$, where $k = 1, \cdots, N$, used in constructing the SVC. Since we obtained only the normalized statevector from the quantum subroutine, an additional machinery is required to estimate its actual magnitude. Therefore, we employ linear regression to estimate both $d$ and $\|\vec{\theta}\|$. Algorithm 2 in Appendix A.2 shows the pseudocode, which was used for construction and validation of the SVC.

## 4 Results

In this section, we discuss the results of our numerical experiments, aiming to evaluate the performance of our VQLS-enhanced QSVM algorithm. In our work, we use the three qubit VQLS model and the size of the kernel matrix is $8 \times 8$.

For the VQLS subroutine, we set the termination condition for the optimization routine as follows: either the program terminates at maximum iterations ($= 300$) or if the cost

value is the same for the last certain number of iterations. For this work,we use the IBM-Q *aer simulator* and the optimizer COBYLA for the classical optimization routine on our local computing resource with a processor Intel Xeon ES-2670, Running at system specifications of 2.60 Hz, 64 GB RAM, and Red Hat Enterprise Linux 7.9.

In Sec. 4.1, we show how employing SVD prior to Pauli decomposition and solving an equivalent problem gives us an edge over merely using Pauli decomposition (Bravo-Prieto et al., 2019), in terms of convergence to a minimum and run-time. In the rest of our analysis, we include SVD as an element in the construction of the classifiers.

In Sec. 4.2 and Sec. 4.3, we use different datasets and different instances within a given dataset to derive SLEs and explore the consequent impact on the convergence of the cost function. This variation leads to SLEs with varying condition numbers, yielding insight into the behavior of VQLS in these cases. In Sec. 4.3, we also analyze the accuracy of classifiers constructed using the VQLS-enhanced QSVM, in comparison to the LS-SVM.

## 4.1 IMPACT OF SVD ON RUN-TIME AND CONVERGENCE

Since VQLS shows promise in terms of scalability to larger systems in Bravo-Prieto et al. (2019), it is crucial to reduce the total number of Pauli strings in Eq. (8) for the computation of the cost function in Eq. (9) and improve its trainability. As proposed in Sec. 3.2, we replace the kernel matrix with its SVD component $\Sigma$. By solving the new system of equations given by Eq. (14), we accelerate the convergence and enhance the trainability compared to the tradional method of using Pauli decomposition for the matrix $A$ in the original problem in Eq. (13).

In our experiment, the number of Pauli strings after decomposition for $A$ and $A_{new}$ are 36 and 8, respectively. Consequently, the total number of expectation values to be computed within the sum in Eqs. (10) and (11) is reduced. For example, when the number of terms in the decomposition is given by $l$, we need $l^2$ loops at most to compute the inner product in Eq. (11). In our case, this translates to 1296 ($36^2$) and 64 ($8^2$) loops for $A$ and $A_{new}$ respectively. This reduction significantly decreases the number of terms required to compute expectation values within Eqs. (10) and (11) and the run-time. The combination of SVD and Pauli decomposition reduces the system run-time to approximately one-sixteenth of what it would be used using the Pauli decomposition alone, when $\langle\psi|\psi\rangle$ and $|\langle b|\psi\rangle|^2$ in Eq. (9) are computed for our specific example. Fig. 7 in Appendix A.5 illustrates the run-time for identifying an optimal set of parameters for the construction of the separating hyperplane when executed using only Pauli decomposition versus the combination of SVD and Pauli decomposition. The cost values start to converge after around 30 min for $A_{new}$ compared to 450 min for $A$ according to the system time. Additionally, the final cost value for $A_{new}$ converged to a notably lower value of 6%, in comparison to the 24% for $A$.

We also note that recasting the problem into Eq. (14) yields a lower minimum of the cost function, indicating a possibly more accurate solution. Fig. 7 in Appendix A.5 also compares the final cost minima. Notably, Bravo-Prieto et al. (2019, Appendix A) discuss precision of the cost function computation and its dependence on sparsity. Specifically, for a $d$-sparse matrix, the discussion presented in Bravo-Prieto et al. (2019) implies that the precision of the cost function computation is inversely proportional to $d$. Consequently, improving sparsity by solving for $A_{new}$ instead of $A$ improves the precision of the cost function calculation. For more details on the role of sparsity in solving SLEs with quantum algorithms, we refer to Harrow et al. (2009) and Childs et al. (2017).

## 4.2 INFLUENCE OF THE CONDITION NUMBER $\kappa$ ON THE CONVERGENCE OF THE COST FUNCTION IN VQLS

We study the influence of parameter $\kappa$ on the convergence of cost function numerically, varying the values of $\kappa$. Numerical experiments are categorized into two parts based on the chosen dataset: toy dataset and the Iris dataset.

### 4.2.1 Results with toy dataset

In this analysis, we randomly choose three different instances of data. The Pauli decomposition of the matrix contains two Pauli strings, III ($\mathbb{1} \otimes \mathbb{1} \otimes \mathbb{1}$) and YYZ ($Y \otimes Y \otimes Z$). Each instance has two different sets of coefficients. Solving each of these SLEs demonstrates a clearer understanding of the impact of $\kappa$ on the convergence of the cost function. Fig. 8 in Appendix A.6 illustrates $\kappa$'s influence on convergence in three instances. It is noteworthy that the results obtained from instances associated with low condition numbers exhibit a better convergence in VQLS.

Given the substantial impact of $\kappa$ on the convergence of the cost function shown in Fig. 8 in Appendix A.6 , we further investigate the relationship between the number of Pauli strings in the decomposition of several matrices with similar condition number and the convergence of the cost function. This analysis involves four instances with the kernel matrix having 10, 15, 20, and 36 Pauli strings. The condition number of all these matrices is $\kappa \approx 3$. The convergence of the cost function is illustrated in Fig. 9 in Appendix A.7. Based on numerical results, the number of Pauli strings does not influence the cost function's convergence.

For SLEs constructed with the toy dataset, VQLS is accurate when the kernel matrix is well conditioned. In such a situation, the number of Pauli strings in its decomposition does not play a major role.

### 4.2.2 Results with the Iris dataset

In this section, we present numerical results that highlight the impact of the condition number $\kappa$ on the convergence of the cost function, when utilizing the Iris dataset to evaluate our approach without the use of SVD. We extracted one instance of training dataset, including seven samples from *Setosa* and *Virginica*, and generated five different kernel matrices using Eq. (6). The condition numbers of these kernel matrices are $\kappa = 5, 10, 19, 144$ and $721$, which is realized by adjusting the hyperparameter $\gamma$ from Eq. (6). The results shown in Fig. 10 in Appendix A.8 align nicely with those for the toy dataset in Sec. 4.2.1 (Fig. 8 in Appendix A.6).

We observe that the use of SVD in preprocessing weakens the existing correlation between the condition number $\kappa$ of the kernel matrix and the convergence of cost function in VQLS. To evaluate our approach's performance when utilizing the SVD, we conducted subsequent experiments using the same five kernel matrices used previously in the analysis presented in Fig. 10 in Appendix A.8. The numerical results demonstrate a lower cost minimum even under a high condition number $\kappa$. This can be observed in Fig. 11 in Appendix A.8.

The weakening of correlation between the condition number and convergence of cost function due to inclusion of SVD is advantageous. This results in a better convergence at higher condition numbers and a significant enhancement in the trainability of variational ansatz.

### 4.3 Performance evaluation of SVC built with VQLS-enhanced QSVM

In this analysis, we consider ten random instances of training sets from the Iris dataset. Four of them have $\kappa \leq 10$, three fall within $10 < \kappa < 100$ and three have $\kappa \geq 100$.

The classification hyperplanes for these ten instances are constructed using the VQLS-enhanced QSVM detailed in Sec. 3. For accuracy validation, we compare the performances of QSVM-based and LS-SVM-based classifiers. The influence of different condition numbers of the kernel matrix, which are manipulated through $\gamma$, is evident on the classifier accuracy as seen in Fig. 2. The final cost values are also plotted for these matrices alongside the accuracy. Furthermore, Table 2 in Appendix A.9 shows the evaluation of classification performance employing a range of metrics.

Furthermore, we repeated each of our numerical experiments five times to examine the stability. Table 4 in Appendix A.10 displays the experimental results for four instances. Based on the table, it is evident that the majority of the outcomes yields similar classification accuracy. It is important to note that having a lower cost value does not inherently guarantee higher classification accuracy. This is due to the fact that a lower cost value does not

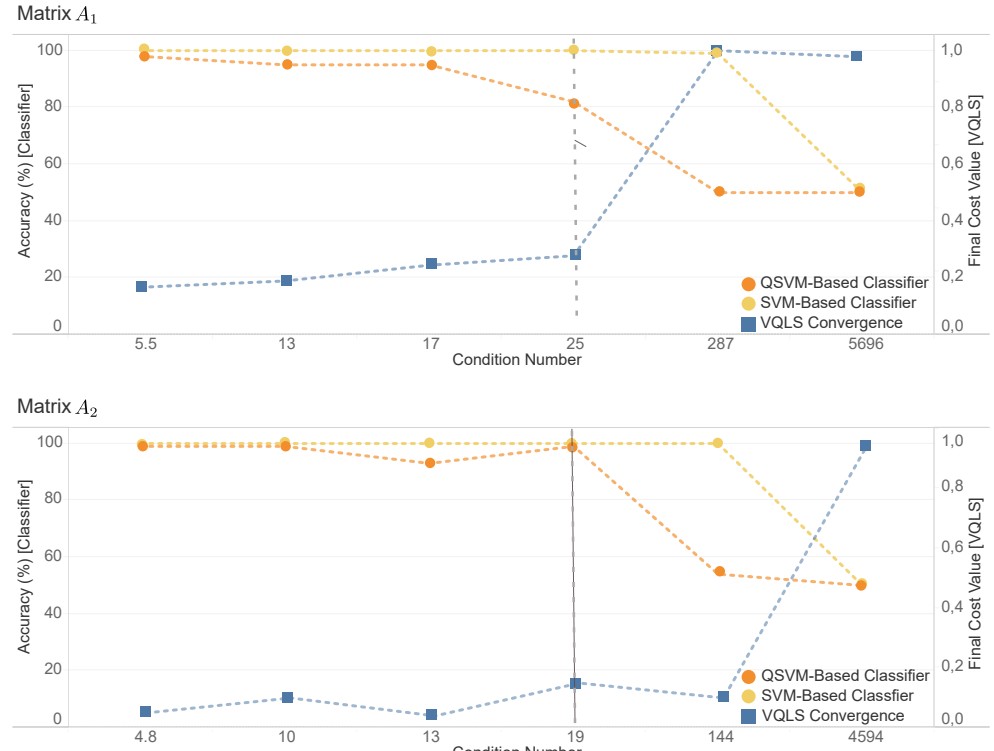

Figure 2: Two instances of classifier accuracy for SLEs with different $\kappa$. On the left side, we compare accuracy of two SVCs on test data, one constructed classically, another from QVSM. It is to be noted that 100% refers to a full correct classification, and the worse classification obtained in this analysis is 50%. On the right, we compare final cost values.

guarantee an accurate solution in the case of VQLS (Bravo-Prieto et al., 2019). Hence, it is important to include a verification step to validate the solution.

## 5 CONCLUSION AND OUTLOOK

This work aims to identify an optimal set of parameters for constructing a classifier on a quantum computer. We then use this classifier to complete the classification tasks in supervised learning. This objective is realized by utilizing our proposed hybrid quantum-classical algorithm on NISQ devices, named as the VQLS-enhanced QSVM. Additionally, we benchmarked this approach by examining the SVC with real-world data, the Iris dataset.

The VQLS-enhanced QSVM is capable of robustly identifying a separating hyperplane that highly accurately classify samples in the test data. We note that SVD is crucial for minimizing the number of controlled unitaries applied during a Hadamard test. Hence, we applied SVD on the kernel matrix $A$ in our numerical experiments. It significantly reduces the number of expectation values computed in one iteration for a faster and more accurate result. Furthermore, appropriately selecting the hyper parameter $\gamma$ in Eq. (6), utilized for the design of the kernel matrix, crucially influences both the trainability of variational ansatze and the classification accuracy. The classifiers constructed using our approach exhibits a strong performance for problems with small condition number of the kernel matrix $A$.

This work can be further explored by employing noise models and executing numerical experiments on real quantum hardware. It is also worthwhile to investigate the scalability of the VQLS-based QSVM with increasing problem size.

## ACKNOWLEDGMENTS

We thank Katja Schladitz and Alexander Geng for helpful conversations regarding the manuscript. This work was funded by the Federal Ministry for Economic Affairs and Climate Action (German: Bundesministerium fÃijr Wirtschaft und Klimaschutz) under the project EniQmA with funding number 01MQ22007A.

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

## A  Appendix

### A.1  Implementation of Quantum circuits

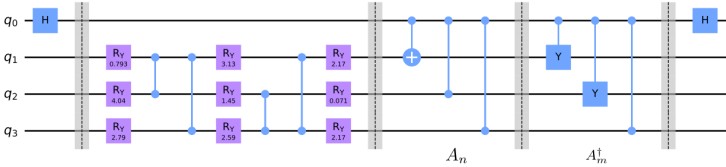

Figure 3: Quantum circuit for the computation of $\langle\psi|\psi\rangle$. The circuit consists of the variational block and is followed by the controlled components.

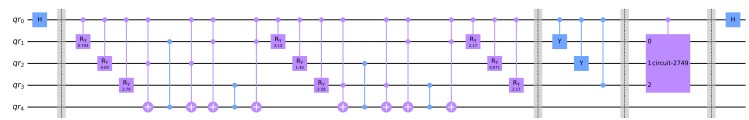

Figure 4: Quantum circuit for the computation of $\langle 0| V(\alpha)^\dagger A_m^\dagger U |0\rangle$. The additional auxiliary qubit is present to facilitate the execution of the CCZ gate.

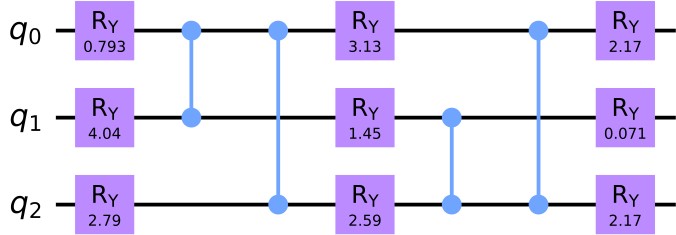

Figure 5: Quantum circuit for hardware efficient ansatz.

## A.2 Pseudocode of VQLS-enhanced QSVM and SVC

### A.2.1 VQLS enhanced QSVM

---

**Algorithm 1** VQLS enhanced QSVM

---

**Input:** Feature samples $X_{train} = \{\vec{x}_1, \cdots, \vec{x}_N\}$ and feature labels $\vec{y}_{train} = \{y_1, \cdots, y_N\}$
**Output:** A set of optimal parameters $\alpha^{opt}$

Normalize $X_{train}$ to $\hat{X}_{train}$ (Eq. (12))
Construct the kernel matrix $K$ (Eq. (6))
Decompose the kernel matrix $K$ into $\sum\limits_{l=0}^{N} c_l A_l$ (Eq. (8))
Initialize iteration $i = 0$, the stop criterion $\epsilon = 0.01$, cost value $C = 1$, the maximum number of iterations $maxIteration = 300$ and initial parameters of parameterized quantum gates $\alpha^i$

$numIteration = 0$
**while** $C > \epsilon$ or $numIteration < maxIteration$ **do**
  $sum1 = 0$
  **for** $A_m$ in $\{A_1, A_2, \cdots, A_N\}$ **do**
    **for** $A_n$ in $\{A_1, A_2, \cdots, A_N\}$ **do**
      Construct the first quantum circuit (Fig. 3)
      Execute the circuit with $shots = 10000$
      Measure the ancillary qubit $q_a$
      Compute $p_{q_a}(|0\rangle) - p_{q_a}(|1\rangle)$ to obtain $\langle 0 | V(\alpha)^\dagger A_m^\dagger A_n V(\alpha) | 0 \rangle$
      $sum1 \mathrel{+}= c_m^* c_n \langle 0 | V(\alpha)^\dagger A_m^\dagger A_n V(\alpha) | 0 \rangle$
    **end for**
  **end for**
  $sum2 = 0$
  **for** $A_m$ in $\{A_1, A_2, \cdots, A_N\}$ **do**
    **for** $A_n$ in $\{A_1, A_2, \cdots, A_N\}$ **do**
      Construct the second quantum circuit for computing the inner product of $\langle 0 | U^\dagger A_n V(\alpha) | 0 \rangle$ (Fig. 4)
      Execute the circuit with $shots = 10000$
      Measure the ancillary qubit $q_a$
      Compute $p_{q_a}(|0\rangle) - p_{q_a}(|1\rangle)$ to obtain the value of $\langle 0 | U^\dagger A_n V(\alpha) | 0 \rangle$
      Again construct the second quantum circuit for computing the inner product of $\langle 0 | V(\alpha)^\dagger A_m^\dagger U | 0 \rangle$
      Execute the circuit with $shots = 10000$
      Measure the ancillary qubit $q_a$
      Compute $p_{q_a}(|0\rangle) - p_{q_a}(|1\rangle)$ to obtain $\langle 0 | V(\alpha)^\dagger A_m^\dagger U | 0 \rangle$
      $sum2 \mathrel{+}= c_m^* c_n \langle 0 | U^\dagger A_n V(\alpha) | 0 \rangle \langle 0 | V(\alpha)^\dagger A_m^\dagger U | 0 \rangle$
    **end for**
  **end for**
  $\frac{|\langle b | \psi \rangle|^2}{\langle \psi | \psi \rangle} \leftarrow \frac{sum1}{sum2}$
  $C \leftarrow 1 - \frac{|\langle b | \psi \rangle|^2}{\langle \psi | \psi \rangle}$
  $i \leftarrow i + 1$
  $numIteration \leftarrow numIteration + 1$
  $\alpha^i \leftarrow$ Update parameters using the optimizer COBYLA
**end while**
**return** $\alpha^{opt}$

---

### A.2.2 SVC

---

**Algorithm 2** SVC

---

**Input:** A set of optimal parameters $\{\alpha_i^{opt}\}$
**Output:** The accuracy of SVC in the validation dataset
   Construct the Hardware-efficient ansatz $V(\alpha)$ (Fig. 5)
   $V(\alpha^{opt}) \leftarrow$ Initialize the Ansatz with the optimal parameters $\{\alpha_i^{opt}\}$
   $|x_{out}\rangle \leftarrow$ Measure all qubits
   $\vec{\theta'} = \frac{\vec{\theta}}{\|\vec{\theta}\|} \leftarrow |x_{out}\rangle$, where $\|\vec{\theta}\|$ is unknown
   $\vec{w'} \leftarrow \sum_{i=1}^{N} \theta_i' \cdot \vec{x}_i$
   $e_i' \leftarrow \frac{\theta_i' \cdot y_i}{\gamma}$
   $\tilde{d}, \|\vec{\theta}\| \leftarrow LinearRegression(\forall i : y_i - y_i e_i' - \vec{w'}^T \vec{x}_i - d = 0)$
   $\vec{\theta} \leftarrow \|\vec{\theta}\| \cdot \vec{\theta'}$
   $\vec{w} \leftarrow \|\vec{\theta}\| \cdot \vec{w'}$
   SVC:
$$\hat{y} = \begin{cases} 1 & \text{if } \vec{w}^T \vec{x} + \tilde{d} \geq 0 \\ -1 & \text{if } \vec{w}^T \vec{x} + \tilde{d} < 0 \end{cases}$$
   **return** $\hat{y}$

---

### A.3 AN INSTANCE OF THE TRAINING DATASET

In this section, in Table 1 we have an overview of one instance of dataset used for training.

Table 1: Overview of one instance of the utilized training dataset.

| No. | Sepal Length | Sepal Width | Petal length | Petal Width | Sepal |
|-----|--------------|-------------|--------------|-------------|-----------|
| 1 | 5.1 | 3.5 | 1.4 | 0.2 | Setosa |
| 2 | 4.9 | 3.0 | 1.4 | 0.2 | Setosa |
| 3 | 4.7 | 3.2 | 1.3 | 0.2 | Setosa |
| 4 | 5.0 | 3.6 | 1.4 | 0.2 | Setosa |
| 5 | 6.7 | 3.0 | 5.2 | 2.3 | Virginica |
| 6 | 6.3 | 2.5 | 5.0 | 1.9 | Virginica |
| 7 | 5.9 | 3.0 | 5.1 | 1.8 | Virginica |

### A.4 INFLUENCE OF THE DATA NORMALIZATION TECHNIQUE ON THE COST FUNCTION CONVERGENCE

Data normalization plays a key role in data preprocessing, particularly in machine learning and data analysis. It encompasses the transformation of data into a standardized format or scale, thereby enhancing its suitability for subsequent analysis or model training. The significance of data normalization is introduced by our cost function convergence analysis in Fig. 6.

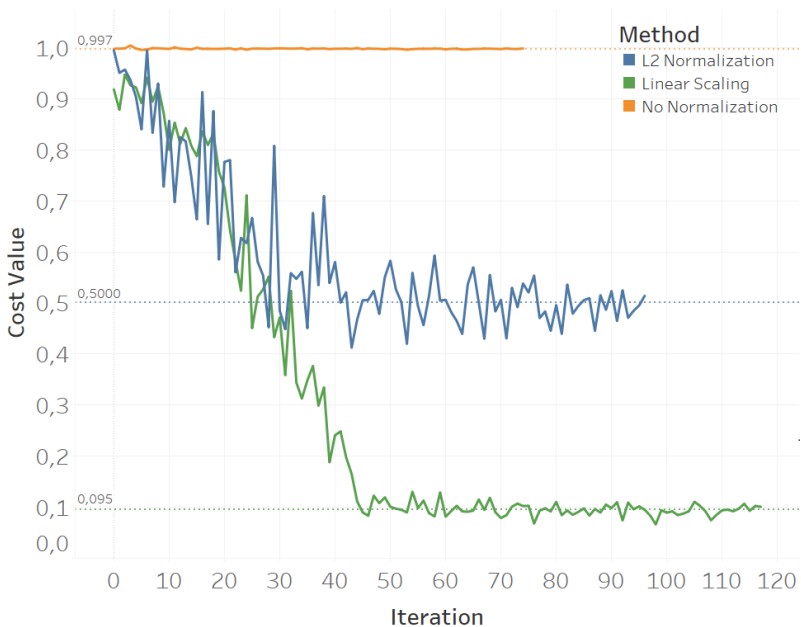

Figure 6: Impact of three data normalization techniques on the cost function convergence in VQLS. It is worth emphasizing that gradient vanishing issues arise when input data is not normalized. Furthermore, linear scaling plays a significant role in mitigating gradient vanishing and facilitates faster and more reliable convergence of the cost function.

## A.5 Run-time Analysis for cost function's convergence

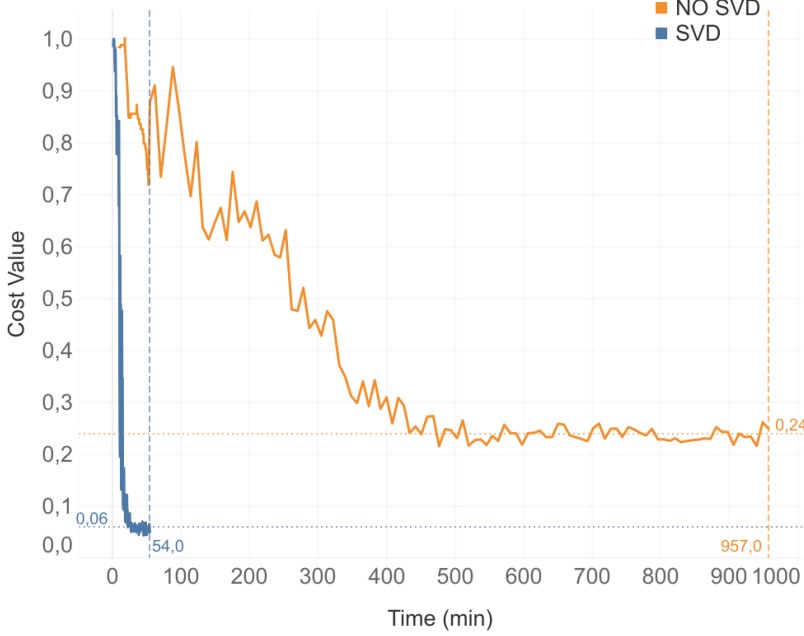

Figure 7: Run-time analysis for the convergence of the cost function for the matrices $A$ and $A_{new}$. The cost values start to converge after around 30 min for $A_{new}$ compared to 450 min for $A$ according to the system time. Additionally, the final cost value for $A_{new}$ converged to a notably lower value of 6%, in comparison to the 24% for $A$.

## A.6  Condition number $\kappa$'s influence on the cost function's convergence

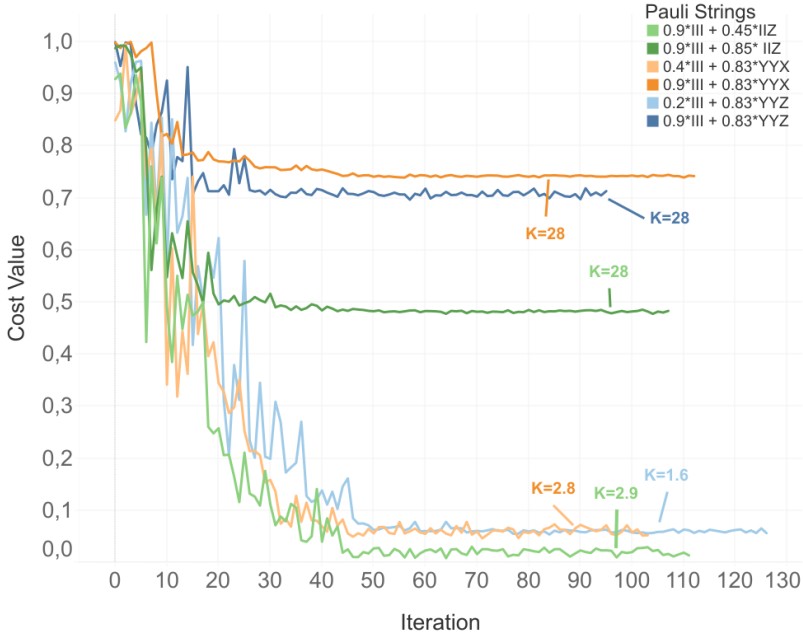

Figure 8: Condition number $\kappa$'s influence on the convergence of the cost function in VQLS. It is noteworthy that the results obtained from instances associated with low condition numbers exhibit a better convergence in VQLS.

## A.7  Influence of the number of Pauli strings

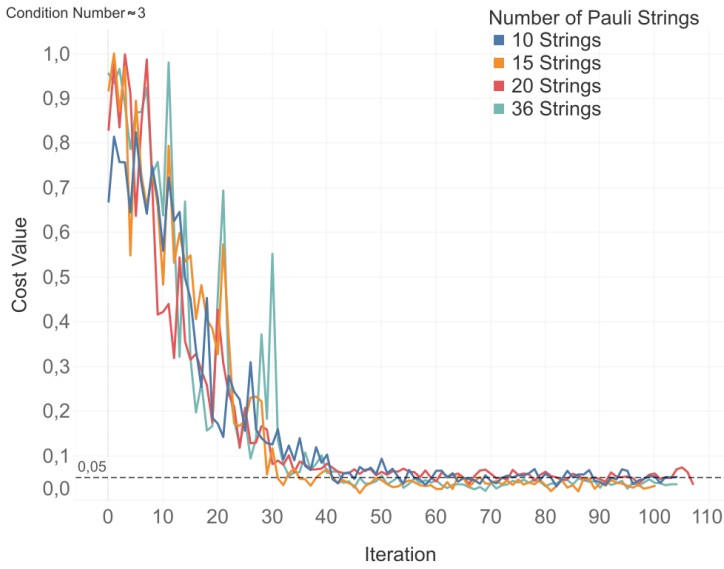

Figure 9: Influence of the number of Pauli strings for a given condition number on the convergence of the cost function in VQLS. It is notable that the number of Pauli strings does not influence the cost function's convergence.

## A.8   IMPACT OF SVD ON THE COST FUNCTION'S CONVERGENCE

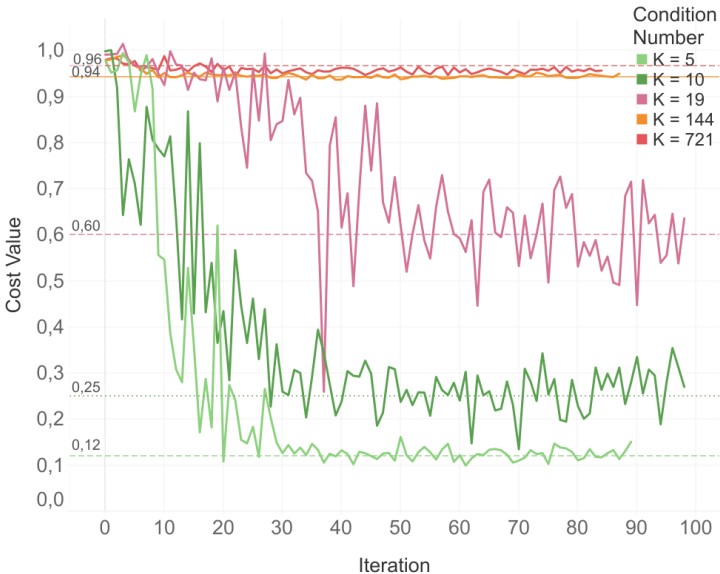

Figure 10:   Impact of $\kappa$ on the convergence of the cost function without the SVD for the Iris dataset .The accuracy of the solution is attributed to lower cost minimum and is better for systems with a lower condition number in VQLS.

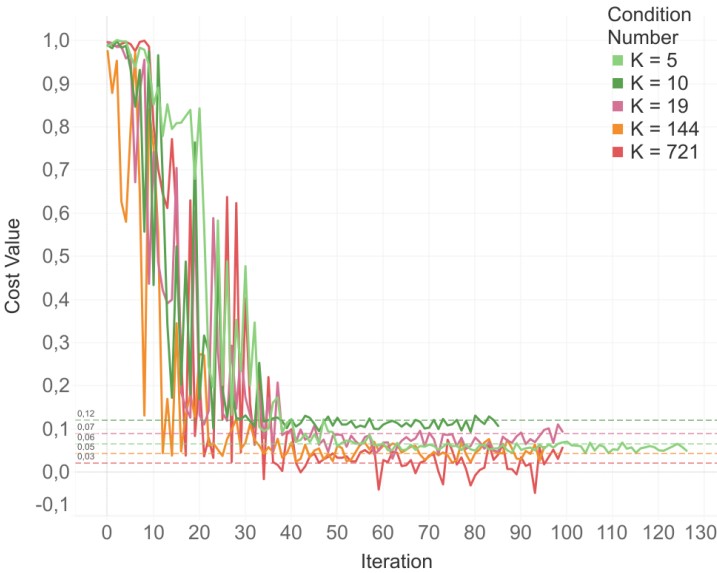

Figure 11: Impact of the condition number $\kappa$ on the convergence with SVD for the Iris dataset. It is worth noting that VQLS demonstrates a notable convergence of cost function in the same instance with the in Fig. 10, even when dealing with matrices featuring a high $\kappa$.

A.9   NUMERICAL RESULTS FOR CLASSIFICATION ACCURACY WITH ADDITIONAL
      INSTANCES OF THE KERNEL MATRIX $A$

Table 2 summarizes the main classification metrics for two instances from the Iris dataset.

Table 2:   A Report showing the main classification metrics for two instances

**Matrix $A_1$**

| $\kappa$ | Class | Precision | Recall | F1-score | Support |
|---|---|---|---|---|---|
| 5.5 | Virginica | 0.98 | 1.00 | 0.99 | 50 |
| | Setosa | 1.00 | 0.98 | 0.99 | 50 |
| 13 | Virginica | 0.91 | 1.00 | 0.95 | 50 |
| | Setosa | 1.00 | 0.90 | 0.95 | 50 |
| 17 | Virginica | 0.91 | 1.00 | 0.95 | 50 |
| | Setosa | 1.00 | 0.90 | 0.95 | 50 |
| 25 | Virginica | 0.74 | 1.00 | 0.85 | 50 |
| | Setosa | 1.00 | 0.64 | 0.78 | 50 |
| 287 | Virginica | 0.00 | 0.00 | 0.00 | 50 |
| | Setosa | 0.50 | 1.00 | 0.67 | 50 |
| 5696 | Virginica | 0.00 | 0.00 | 0.00 | 50 |
| | Setosa | 0.50 | 1.00 | 0.67 | 50 |

**Matrix $A_2$**

| $\kappa$ | Class | Precision | Recall | F1 - score | Support |
|---|---|---|---|---|---|
| 4.8 | Virginica | 0.98 | 1.00 | 0.99 | 50 |
| | Setosa | 1.00 | 0.98 | 0.99 | 50 |
| 10 | Virginica | 0.98 | 1.00 | 0.99 | 50 |
| | Setosa | 1.00 | 0.98 | 0.99 | 50 |
| 13 | Virginica | 0.88 | 1.00 | 0.93 | 50 |
| | Setosa | 1.00 | 0.86 | 0.92 | 50 |
| 19 | Virginica | 0.98 | 1.00 | 0.99 | 50 |
| | Setosa | 1.00 | 0.98 | 0.99 | 50 |
| 144 | Virginica | 0.52 | 1.00 | 0.68 | 50 |
| | Setosa | 1.00 | 0.08 | 0.15 | 50 |
| 4594 | Virginica | 0.00 | 0.00 | 0.00 | 50 |
| | Setosa | 0.50 | 1.00 | 0.67 | 50 |

we present the classification accuracy for the repetitions of a specific instance in Table 3. This also serves as a stability analysis for the program.

Table 3: Analysis of the stability of SVC constructed by the VQLS-enhanced QSVM in one instance

| Instance | $\kappa$ | No. of incorrect classification | accuracy of our SVC | accuracy of classical SVC |
|---|---|---|---|---|
| | 5 | 1 | 99% | 100% |
| | 11 | 1 | 99% | 100% |
| 1 | 19 | 7 | 93% | 100% |
| | 144 | 46 | 54% | 100% |
| | 4594 | 50 | 50% | 50% |
| | 6 | 2 | 99% | 100% |
| | 13 | 5 | 94% | 100% |
| 3 | 25 | 5 | 50% | 100% |
| | 287 | 18 | 50% | 99% |
| | 5696 | 1 | 50% | 50% |
| | 18 | 1 | 99% | 100% |
| | 30 | 46 | 54% | 100% |
| 4 | 319 | 3 | 97% | 50% |
| | 635 | 47 | 53% | 47% |
| | 6961 | 1 | 99% | 50% |
| | 5 | 1 | 99% | 100% |
| | 11 | 6 | 94% | 100% |
| 5 | 21 | 50 | 50% | 100% |
| | 138 | 1 | 99% | 100% |
| | 5230 | 1 | 99% | 50% |
| | 18 | 1 | 99% | 100% |
| | 30 | 46 | 54% | 100% |
| 6 | 319 | 3 | 97% | 50% |
| | 635 | 47 | 53% | 47% |
| | 6961 | 1 | 99% | 50% |
| | 21 | 50 | 50% | 100% |
| | 50 | 18 | 82% | 100% |
| 7 | 76 | 37 | 63% | 77% |
| | 178 | 1 | 99% | 59% |
| | 8302 | 50 | 50% | 50% |
| | 22 | 48 | 52% | 100% |
| | 30 | 49 | 51% | 100% |
| 8 | 102 | 45 | 55% | 61% |
| | 156 | 1 | 99% | 90% |
| | 7880 | 50 | 50% | 50% |
| | 26 | 50 | 50% | 100% |
| | 34 | 1 | 99% | 100% |
| 9 | 47 | 1 | 99% | 81% |
| | 544 | 1 | 99% | 50% |
| | 7528 | 1 | 99% | 50% |
| | 25 | 50 | 50% | 100% |
| | 34 | 49 | 51% | 100% |
| 10 | 83 | 50 | 50% | 100% |
| | 178 | 50 | 50% | 100% |
| | 8936 | 50 | 50% | 50% |

A.10 NUMERICAL RESULTS FOR EVALUATING THE STABILITY WITH ADDITIONAL INSTANCES OF THE KERNEL MATRIX $A$

The data in Table 4 indicates that most of the classification results are similarly precise.

Table 4: Analysis of the stability of SVC constructed by the VQLS-enhanced QSVM in one instance

*Instance 1*

| $\kappa$ | No. | No. of incorrect classification | accuracy of our SVC | accuracy of classical SVC |
|---|---|---|---|---|
| 4.8 | 1 | 1 | 99% | |
| | 2 | 1 | 99% | |
| | 3 | 1 | 99% | 100% |
| | 4 | 1 | 99% | |
| | 5 | 41 | 59% | |
| 287 | 1 | 1 | 99% | |
| | 2 | 1 | 99% | |
| | 3 | 1 | 99% | 100% |
| | 4 | 1 | 99% | |
| | 5 | 46 | 54% | |
| 4594 | 1 | 50 | 50% | |
| | 2 | 50 | 50% | |
| | 3 | 50 | 50% | 50% |
| | 4 | 50 | 50% | |
| | 5 | 48 | 52% | |

*Instance 2*

| $\kappa$ | No. | No. of incorrect classification | accuracy of our SVC | accuracy of classical SVC |
|---|---|---|---|---|
| 17 | 1 | 1 | 99% | |
| | 2 | 3 | 97% | |
| | 3 | 1 | 99% | 100% |
| | 4 | 1 | 99% | |
| | 5 | 37 | 63% | |
| 30 | 1 | 1 | 99% | |
| | 2 | 49 | 51% | |
| | 3 | 1 | 99% | 100% |
| | 4 | 1 | 99% | |
| | 5 | 5 | 95% | |
| 319 | 1 | 1 | 99% | |
| | 2 | 3 | 97% | |
| | 3 | 1 | 99% | 100% |
| | 4 | 14 | 86% | |
| | 5 | 48 | 52% | |

### Instance 3

| $\kappa$ | No. | No. of incorrect classification | accuracy of our SVC | accuracy of classical SVC |
|---|---|---|---|---|
| | 1 | 1 | 99% | |
| | 2 | 7 | 93% | |
| 11 | 3 | 1 | 99% | 100% |
| | 4 | 44 | 56% | |
| | 5 | 41 | 59% | |
| | 1 | 14 | 86% | |
| | 2 | 50 | 50% | |
| 35 | 3 | 1 | 99% | 100% |
| | 4 | 1 | 99% | |
| | 5 | 2 | 98% | |
| | 1 | 2 | 98% | |
| | 2 | 31 | 69% | |
| 14742 | 3 | 37 | 63% | 50% |
| | 4 | 1 | 99% | |
| | 5 | 49 | 51% | |

### Instance 4

| $\kappa$ | No. | No. of incorrect classification | accuracy of our SVC | accuracy of classical SVC |
|---|---|---|---|---|
| | 1 | 50 | 50% | |
| | 2 | 50 | 50% | |
| 8 | 3 | 46 | 54% | 100% |
| | 4 | 49 | 51% | |
| | 5 | 49 | 51% | |
| | 1 | 1 | 99% | |
| | 2 | 50 | 50% | |
| 21 | 3 | 48 | 52% | 100% |
| | 4 | 1 | 99% | |
| | 5 | 46 | 54% | |
| | 1 | 50 | 50% | |
| | 2 | 50 | 50% | |
| 222 | 3 | 1 | 99% | 84% |
| | 4 | 50 | 50% | |
| | 5 | 50 | 50% | |

