# OpenReview forum: "Variational Quantum Linear Solver enhanced Quantum Support Vector machine"
_ICLR.cc/2024/Conference — Submitted to ICLR 2024_

### Official Review · Reviewer_qfy2 · 2023-10-24

**Soundness:** 2 fair
**Presentation:** 2 fair
**Contribution:** 2 fair
**Rating:** 1
**Confidence:** 5

**Summary:**

The authors propose a so-called variational quantum linear solver (VQLS) enhanced quantum support vector machine (QSVM) to determine the hyperplane of classification problems. The essence of the proposed algorithm is to utilize the Hardware Efficient Ansatz (HEA) to solve classification problems. Numerical results on the tailored Iris dataset are provided.

**Strengths:**

The overall structure of this paper is ok.

**Weaknesses:**

1. The proposed method is too simple and outdated. The essence of this paper is utilizing HEA to solve classification problems which are widely studied in the previous literature [1,2]. Using HEA to solve such a problem is quite trivial and there seems no supremacy in using HEA to solve this kind of problem. There is also literature using Quantum Architecture Search to design problem-specific ansatz for classification problems [3,4].
2. This paper lacks a theoretical analysis of the expressivity and the trainability of the proposed quantum ansatz.
3. The tailored Iris dataset is too simple. The authors only conduct experiments with 3 qubits and a very small kernel, which seems no big difference from the toy dataset. The authors constantly mention that the main advancement of this paper is that they use real-world datasets, but results at this level are clearly not enough (with much larger scaling on MNIST [2]).
4. Poor arrangement with lots of important information in the appendix, leading to difficulty in understanding the paper.
5. **Including acknowledgment in the paper is a clear violation of the double-blind rule of ICLR which should be desk rejected.**



[1] Quantum convolution neural networks

[2]  Quantum convolutional neural network for classical data classification.

[3] QuantumDARTS: Differentiable Quantum Architecture Search for Variational Quantum Algorithms

[4] Quantum circuit architecture search for variational quantum algorithms

**Questions:**

I have no further questions.

**Details Of Ethics Concerns:**

acknowledgment involved

---

> ### Author Response · Authors · 2023-11-11
> **Perspectives on the comment from Reviewer qfy2 and expectation for your suggestions at two points (edited with italic fonts)**
>
> We highly value the reviewer's constructive feedback and would like to share our perspectives on the highlighted issues:
>
>
> **To Point One:**
> The performance and application of HEA have been extensively discussed in these literature. I concur with these findings.
>
> In our research, we noticed that VQLS  demonstrates notable scalability on NISQ- devices for handling larger problem sizes with toy examples, as detailed in [1]. After a thorough review of papers citing VQLS's original research, a key gap we observed was the lack of exploration into VQLS's performance in real-world scenarios. This paper aims to bridge this gap, exploring an area previously uncharted in quantum computing research.
>
> By utilizing HEA into our algorithm, we have effectively tackled implementation challenges, demonstrating the practicality and feasibility of QSVM in the context of NISQ technology. Our results provide valuable insights into the practical applications and scalability of VQLS, offering a new perspective to those interested in the field.
>
> While it appears that HEA doesn't necessarily offer superiority in solving such problems, our research underscores the importance of continual exploration and innovation. Each new study, including ours, brings us closer to realizing the full potential of VQLS theoretically and numerically.
>
> **To Point Two:**
>
> Having established the feasibility of our algorithm, we plan to employ these two descriptors to evaluate the performance of various Ansätze. Your insightful comments have inspired us to further investigate how these descriptors impact the performance of our algorithm across different Ansätze.
>
> **To Point Three:**
>
> Our approach differs notably from the use of toy datasets, which often select specific cases to demonstrate their concepts.
>
> The HHL-based QSVM was initially implemented as described in [2]. However, its kernel  must satisfy this limitation: the condition number of the kernel matrix needs to be scaled to polylog(N), where N represents the size of the linear system, as detailed in [3]. In contrast, our experiments involved the random selection of seven training samples applied to three qubits.  This critical distinction that we believe is important to highlight.
>
> Furthermore, after having submitted our paper, *we expanded our evaluation to include additional datasets concerning heart disease and TIG aluminium 5083 from Kaggle, also implemented on three qubits. When these results affirmatively demonstrate the performance of our approach, we are ready to update our paper with these findings upon receiving your feedback.*
>
> As for the scalability of our algorithm, we are currently reviewing literature on it and strategies to address the challenge of the barren plateau. Following this review, we plan to further evaluate and refine our algorithm in the subsequent phase of our research.
>
>
> **To Point Four:**
> I apologize if the current arrangement of our paper has hindered your understanding. Due to page constraints, we placed essential numerical results in the appendix. *If you have any suggestions for improving this aspect, I would greatly appreciate your input and will promptly revise the paper to enhance its structure*.
>
> **TO Point Five:**
> Regarding the 'double blind reviewing' process for ICLR 2024, I revisited the guidelines but did not findthis specific detail yet. However, after receiving your feedback, I searched on relevant information and found this specific detail in the ICLR 2023 guidelines. I apologize for any rule infringement due to my lack of awareness and am grateful for your pointing out this important aspect. I will ensure to take this into consideration for future submissions that involve double blind reviewing
>
> Thank you once again for your valuable feedback. We eagerly await your response and, upon receiving it, will diligently work on refining this paper according to your valuable suggestions.
>
>
> ---
>
> [1] Bravo-Prieto, Carlos, et al. "Variational quantum linear solver." arXiv preprint arXiv:1909.05820 (2019).
>
> [2]  Zhaokai Li, Xiaomei Liu, Nanyang Xu, and Jiangfeng Du. Experimental realization of a quantum support vector machine. Physical review letters, 114(14): 140504, 2015.
>
> [3] Liu, Xiaonan, et al. "Survey on the Improvement and Application of HHL Algorithm." _Journal of Physics: Conference Series_. Vol. 2333. No. 1. IOP Publishing, 2022.

---

> > ### Comment · Reviewer_qfy2 · 2023-11-19
> >
> > After reading the rebuttal, it is quite clear that this paper has certain weaknesses and limitations that could not be overcome easily. As far as I'm concerned, this paper does not meet the standard of ICLR  from the perspectives of novelty, soundness, and contribution. The rating is partially caused by the ethics problem and I will leave the issue to AC.

---

### Official Review · Reviewer_aeDd · 2023-10-29

**Soundness:** 2 fair
**Presentation:** 3 good
**Contribution:** 2 fair
**Rating:** 5
**Confidence:** 3

**Summary:**

The authors are motivated by the possibility of applying large-scale quantum support vector machines (QSVM) to real-word problems on near-term, noisy quantum hardware. They propose to use variational quantum linear solver (VQLS) to substitute expensive matrix inverse (done with HHL) in the original approach.

The submission is an interesting but very preliminary work on implementing QSVM using NISQ processors.

**Strengths:**

The authors assess their approach through a set of numerical experiments using the Iris dataset, which comprises three distinct iris plant species. They evaluate the algorithm's effectiveness by building a classifier capable of handling feature spaces ranging from one to seven dimensions.

Both classical and quantum computing are harnessed by the authors for various parts of their algorithm, effectively mitigating implementation challenges. These improvements include enhancing the trainability of the variational ansatz and reducing runtime for cost calculations. Based on the numerical experiments, the authors' approach demonstrates the ability to identify a separating hyperplane in an 8-dimensional feature space and consistently delivers strong performance across different instances of the same dataset.

**Weaknesses:**

The authors do not present any new analytical studies to support their conclusions. The work presented by the authors is entirely numerical. It is not known how the approach will scale with the number of qubits required in the setup. The paper presents only very small numerical examples (3-4 qubits). The method works in that regime but it is not clear how it will behave for qubit counts required to utilize quantum advantage. To remedy the situation, the authors could present large-scale numerical examples or attempt to derive analytical result that would give an argument for favorable scaling.

The authors noted that matrix A may, in general, require exponentially many terms in the decomposition in Eq. (2). [As a side note, they incorrectly called it an eigen decomposition.] That may be a problem that ruins the entire approach. The authors did not present a scalable approach to solve that issue. They propose to perform SVD on the matrix A that improve their results. This is not a scalable solution and introduces other problems such as finding circuits that perform unitary evolution given by V and W in Eq. (5).

Overall, it is hard to be convinced by their results that their work “takes us one step closer to realizing possible practical applications of QSVM on a quantum computer in the NISQ-era”.

**Questions:**

- The authors could present large-scale numerical examples that would give an argument for favorable scaling under realistic noise models.
- The authors could attempt to derive analytical results to support the feasibility of running QSVM in the NISQ era.

---

> ### Author Response · Authors · 2023-11-11
> **Response to Reviewer aeDd's Comments and Anticipation of Further Feedback**
>
> Thank you very much for your thorough review and the insightful questions regarding our algorithm. I am pleased to provide answers to your questions:
>
> **Analytical results to support our conclusion - the feasibility of the VQLS-based QSVM in the NISQ-era:**
>
> In our study, we employed a variational method that facilitates task-oriented programming and a Hardware-Efficient Ansatz. This approach is specifically designed to accommodate encoding symmetries and bring correlated qubits into closer proximity, thus reducing circuit depth [1]. A key benefit of this strategy is the maintenance of shallow quantum circuit depth, which is critical for mitigating noise— a notable contrast to algorithms developed for fault-tolerant quantum computing [2]. This framework is a crucial factor making our VQLS-based QSVM feasible on NISQ computers.
>
> Another important aspect is the density matrix of the unitaries in the decomposition of matrix $A$, as detailed in Eq. A 4 of [3]. This matrix has a prefactor proportional to $d$ , indicating a $d$ -sparse matrix structure. The precision in estimating the expectation value of each unitary is inversely proportional to $d$. We conducted a comparative analysis of our algorithm's convergence in two scenarios:  : 1) without employing SVD, and 2) with the application of SVD. The empirical results, as illustrated in Figure 10 of our paper, highlight improved trainability of VQLS-based QSVM when SVD is applied before encoding matrix $A$ into the quantum computer. This suggests that the application of SVD effectively reduces the sparsity of the Hamiltonian, thereby enhancing its trainability.
>
> **Regarding to the large-scaled numerical results**
>
> VQLS has shown significant scalability on NISQ- device when addressing larger problem sizes with artificial examples, as elaborated in [3]. This discovery underpins our initial step in proving the feasibility of VQLS-base QSVM in the real-world application.
>
> Moving forward, our research will delve into scalability aspects with large-scale qubits, addressing potential challenges and offering analytical insights. We are currently examining literature on variational algorithms' challenges, such as the barren plateau phenomenon, particularly in contexts involving a large number of qubits.
>
> A thorough investigation, mitigation of challenges, and analysis of the numerical results for large-scale scenarios will necessitate more time. If feasible, I am keen to include these findings in the current paper. I hope for your understanding and thank for your suggestion and support in this matter.
>
> **How it will behave for qubit counts required to utilize quantum advantage**
>
> Quantum computing is able to encode an exponentially increasing amount of data into a linearly growing number of qubits[1]. In our case, we encoded seven examples into three qubits, as mentioned in your comment. However, in the current NISQ era, variational algorithms do not yet demonstrate a clear quantum advantage, as the optimization training process is NP-hard [4]. Should we achieve the development of noise-tolerant quantum computers, it would be possible to directly implement QSVM as proposed in [5]. This advancement could significantly reduce the time complexity from  $\mathcal{O}(\log(\epsilon ^{-1})poly(N, M))$  in classical SVM to  $\mathcal{O}(\log NM)$  [5], marking a substantial leap in computational efficiency.
>
>
> *I would like to confirm whether my response has adequately addressed the concern you raised in the 'weaknesses' section. If so, I am eager to receive your feedback on whether this content shall be added in the paper. If my response did not fully address your point, I would appreciate more detailed feedback to help me refine and improve it accordingly*.
>
>
> **One Question:**
> *Regarding Equation 2 in my paper, which details the separating hyperplane, could you please provide more specific information or clarification on the issue you've identified with this equation in the "weaknesses" section?*
>
>
> ---
>
> [1]  M. Cerezo _et al._, “Variational quantum algorithms,” _Nat Rev Phys_, vol. 3, no. 9, pp. 625–644, Aug. 2021, doi: [10.1038/s42254-021-00348-9]
>
> [2]  J. Tilly _et al._, “The Variational Quantum Eigensolver: A review of methods and best practices,” _Physics Reports_, vol. 986, pp. 1–128, Nov. 2022, doi: [10.1016/j.physrep.2022.08.003]
>
> [3] Bravo-Prieto, Carlos, et al. "Variational quantum linear solver." arXiv preprint arXiv:1909.05820 (2019).
>
> [4]  Lennart Bittel, and Martin Kliesch. Training Variational Quantum Algorithms Is NP-Hard. Physical review letters, 127 (12): 120502, 2021.
>
> [5]  P. Rebentrost, M. Mohseni, and S. Lloyd, “Quantum support vector machine for big data classification,” _Phys. Rev. Lett._, vol. 113, no. 13, p. 130503, Sep. 2014, doi: [10.1103/PhysRevLett.113.130503]

---

### Official Review · Reviewer_w8ed · 2023-10-31

**Soundness:** 1 poor
**Presentation:** 2 fair
**Contribution:** 2 fair
**Rating:** 3
**Confidence:** 4

**Summary:**

The paper proposes to leverage the variational quantum linear system solver (VQLS) in the quantum support vector machines (QSVM), called VQLS-enhanced QSVM. It uses VQLS to obtain a solution to a least-squares problem, which represents a separating hyperplane in QSVM. The design makes use of the advantage of variational algorithms on NISQ devices. The paper then provides an empirical case study on noiseless quantum simulators with a classification dataset in computer vision and a toy dataset. The authors employ a classical SVD process before running the VQLS-enhanced QSVM to reduce the sparsity of the Hamiltonian and improve the performance.

**Strengths:**

The use of SVD to regularize the data is clever and shows good performance.
The experiments are described in detail, demonstrating the impact of different subroutines of the proposed procedure.

**Weaknesses:**

The motivation for using VQLS to obtain a solution to the hyperplane is to have a robust solution on NISQ devices where noises play the main role and obstruct the original HHL subroutine in the QSVM. However, theoretical analysis and experiments in the paper do not provide enough evidence to support the fact that the proposed method mitigates the effects of noise. The theoretical analysis only considers perfect implementation and the experiments are conducted on an ideal circuit simulator. The suggested robustness of the method might be severely limited in the subroutines requiring precise control of the quantum system, e.g., the Hadamard test. The evidence in the paper does not convince me of the necessity of substituting VQLS for HHL in the QSVM in the first place.

The use of SVD to regularize the data seems not scalable. One of the benefits of QSVM is the ability to deal with high-dimensional data, while the classical SVD regularization can only be performed in small cases. This fact limits the scenarios where SVD might be useful.

The comparison with other quantum machine learning methods is also minimally discussed. The VQLS-enhanced QSVM is compared with only QSVM and SVM in only one example, where the performance of the VQLS-enhanced version seems much poorer than the other methods when the condition number is small.

The SVD regularization is also applicable to QSVM and SVM, while in the experiments it seems that SVD is not applied to the other methods. I believe these are essential to have a fair comparison of the methods.

The lack of consideration of noises also weakens the claims on the effectiveness of SVD. Are the noises affecting the performance of SVD?

Minor typos:
Page 1, "provided as an *imput* to the quantum hardware"
Page 7, "IBM-Q aer simulator" => IBM-Q Aer simulator
Page 7, ", Running at"

**Questions:**

According to the weakness, several questions are poised:
* How do noises affect the performance of VQLS-enhanced QSVM and HHL-based QSVM?
* How scalable is the SVD regularization?
* What are the performance of other methods in the experiments, with SVD regularization applied?
* How do noises affect the performance of SVD?

---

> ### Author Response · Authors · 2023-11-11
> **Response to Reviewer w8ed's Comments and Awaiting Further Feedback**
>
> I am grateful for your insightful comment. The mino types will be corrected in the next version of paper. Please find below my detailed response, reflecting my best effort to comprehensively address your questions:
>
>
> **Noise affect the performances of VQLS-enhanced QSVM and HHL-based QSVM**
>
> The fidelity of experimental results in quantum systems, particularly in the context of numerical experiments, has been investigated as detailed in [1]. To the best of our knowledge, the impact of noise on HHL-based QSVM remains unexplored. While there is an experimental implementation of the HHL-based QSVM algorithm presented in [2], the effects of noise in real hardware environments have not been thoroughly examined in [2].
>
> Addressing noise in quantum hardware is a substantial challenge in quantum information theory, with topics like quantum noise correction and mitigation being pivotal areas of research. However, it's important to mention that our current work does not delve into the  effects of noise on quantum algorithms or the strategies for mitigating such noise.
>
> Concurrently, our paper seeks to address a significant research gap: the exploration of VQLS' feasibility in practical, real-world scenarios. This gap was identified after an extensive review of existing literature citing the original research on VQLS in [3]. By focusing on this uncharted area in quantum computing research, we aim to contribute valuable insights and advancements in the field.
>
>
> Thank you for your constructive feedback. We acknowledge the importance of this topic. The investigation into noise effects in quantum simulations and the development of strategies to mitigate these effects will be a focus in our future work. We also hope that this paper will inspire other researchers to further investigate and contribute to this area.
>
>
> **How scalable is the SVD regularization?**
>
> The time complexity of SVD is $O(min ( m^{2}n, mn^{2} ))$, as detailed in [4]. The underlying intuition for applying SVD to matrix $A$ is similar to the intuition in the hybrid classical-quantum variational algorithms. In such algorithms, A shallow depth of quantum ansatze is implemented and processed on a quantum computer, effectively reducing the depth of the quantum circuit [5]. However, this benefit comes at the cost of increased time complexity for solution finding in the classical computer.
>
>
> In the NISQ era, striking a balance between computational resources allocated to quantum and classical computers is pivotal. Our comparative analysis of results with and without SVD application, as presented in Figure 10 in our paper, empirically substantiates the theoretical conclusions drawn in Appendix 1 of [3]. This comparison shows that while SVD regularization increases computational complexity on the classical computers, it offers significant benefits in terms of improving algorithm performance.
>
> **The performance of other methods in the experiments, with SVD regularization applied**
>
> To the best of our knowledge, our paper could be the first to apply VQLS in the realm of QSVM for classifying on the real-world datasets, as demonstrated through our numerical experiments. Moreover, we investigated the feasibility of this algorithm.
>
>
> **How do noises affect the performance of SVD?**
> The performance of SVD is computed on the classical computer in the data preparation. The noisy data affected the performance of SVD, as detailed in [6],[7].
>
>
> ---
> [1]  Liu, Xiaonan, et al. "Survey on the Improvement and Application of HHL Algorithm." _Journal of Physics: Conference Series_. Vol. 2333. No. 1. IOP Publishing, 2022.
>
> [2]  Zhaokai Li, Xiaomei Liu, Nanyang Xu, and Jiangfeng Du. Experimental realization of a quantum support vector machine. Physical review letters, 114(14): 140504, 2015.
>
> [3] Bravo-Prieto, Carlos, et al. "Variational quantum linear solver." arXiv preprint arXiv:1909.05820 (2019).
>
> [4]  N. J. Z. Mamat and J. K. Daniel, ‘Statistical analyses on time complexity and rank consistency between singular value decomposition and the duality approach in AHP: A case study of faculty member selection’, _Mathematical and Computer Modelling_, vol. 46, no. 7, pp. 1099–1106, 2007.
>
>
> [5] M. Cerezo _et al._, “Variational quantum algorithms,” _Nat Rev Phys_, vol. 3, no. 9, pp. 625–644, Aug. 2021, doi: [10.1038/s42254-021-00348-9]
>
> [6]  De Moor, Bart. "The singular value decomposition and long and short spaces of noisy matrices." _IEEE transactions on signal processing_ 41.9 (1993): 2826-2838.
>
> [7] Henry, E. R., and J. Hofrichter. "[8] Singular value decomposition: Application to analysis of experimental data." _Methods in enzymology_. Vol. 210. Academic Press, 1992. 129-192.

---

> > ### Comment · Reviewer_w8ed · 2023-11-21
> >
> > Thanks for the response. I do not have further comments.

---

### Official Review · Reviewer_NGt1 · 2023-11-01

**Soundness:** 3 good
**Presentation:** 3 good
**Contribution:** 2 fair
**Rating:** 3
**Confidence:** 4

**Summary:**

This paper proposes a new quantum support vector machine (QSVM) based on a variational quantum linear system solver. This method formulates the SVM as a linear programming which is equivalent to solving a system of linear equations (Chua 2003, Robentrost et al. 2014). Then, the system of linear equations is solved with a variational quantum linear system solver due to Bravo-Prieto et al. (2019). The performance of this method is demonstrated on noise-free IBM-Q simulators with 3 qubits (so the dimension of the feature space is $2^3=8$). The numerical experiment uses the Iris dataset. The numerical results show that this new QSVM serves as a good classifier when the condition number of the kernel matrix is small.

**Strengths:**

The formulation of this work seems reasonable and the preliminaries are written clearly. In previous works (Havlicek et al. 2019, Li et al. 2022, Zhang et al. 2022, Ezawa et al. 2022), a similar variational approach was considered but the numerical experiments were conducted to solve a toy-model problem with only two features. This work conducted numerical experiments with a slightly larger problem (8-dimensional feature space) from the Iris dataset.

**Weaknesses:**

The method proposed in this paper appears to be a simple combination of two existing subroutines (the least square formulation of SVM and its equivalence to solving linear systems & variational quantum linear system solvers), which does not seem novel. To implement the variational quantum linear system solver, the authors apply SVD to the feature matrix $A$ to get a simpler Pauli decomposition -- this approach seems to be *ad hoc* and not scalable to higher dimensions. The numerical experiment is small (the experiment was simulated using noise-less IBM-Q simulators, not with real quantum hardware) and the results are weak (this new method is not as good as classical ones when the condition number is large).

**Questions:**

To achieve quantum speedup using a variational quantum linear system solver, the matrix $A$ must be represented as a sum of Pauli (or more generally, unitary) operators. However, in the setting of this paper, the input data (i.e., the matrix $A$) is given as a classical matrix. For ease of implementation, the authors used SVD to get a simple Pauli decomposition of $A$. Using classical SVD to pre-process the data is not scalable, as the cost grows superlinearly with the size of the matrix $A$ (the quantum speedup comes from the assumption that we can represent the matrix $A$ using poly-log quantum resources). In other words, without an efficient data-loading procedure, I do not think the method proposed in this paper would achieve a significant end-to-end quantum advantage on a real NISQ device.

---

> ### Author Response · Authors · 2023-11-12
> **Responding to Reviewer NGt1's Remarks and Looking Forward to Further Insights**
>
> Thank you for your thoughtful and insightful comment. I am eager to discuss our viewpoint on the issues you've highlighted and share our perspective.
>
> **Regarding the quantum advantage:**
>
> In the present era of NISQ technology, variational algorithms have yet to demonstrate a definitive quantum advantage, primarily due to the NP-hard nature of the optimization training process, as noted in [1]. However, the potential for significant breakthroughs exists with the development of noise-resistant quantum computers and quantum RAM. Such advancements would enable the direct implementation of QSVM, as proposed in [2]. This implementation could drastically reduce the time complexity from the classical SVM's    $\mathcal{O}(\log(\epsilon ^{-1})poly(N, M))$ to $\mathcal{O}(\log NM)$ [2],   representing a remarkable improvement in computational efficiency. The research community is increasingly focused on harnessing quantum advantage through the ongoing development and refinement of general quantum algorithms. We hope that this paper will motivate other researchers to further investigate and contribute to the quest for achieving quantum advantage.
>
>
> **Regarding the intuition of applying SVD:**
>
> The intuition behind applying SVD to matrix $A$ parallels the logic of hybrid classical-quantum variational algorithms. In these algorithms, the quantum component can be executed on NISQ-era computers by reducing the depth of the quantum circuit, as discussed in [3]. However, this reduction in quantum complexity comes at the cost of increased time complexity for finding the solution. This highlights the critical need for a balance of computational resources between quantum and classical computers during the NISQ era.
>
> Our study, particularly the comparative analysis showcased in Figure 10 of our paper, empirically validates the theoretical analysis drawn in Appendix 1 of [4]. The results, comparing scenarios with and without the use of SVD, demonstrate that although SVD introduces greater computational complexity on the classical side, it significantly enhances the overall performance of the algorithm. This finding underscores the importance of strategically integrating classical and quantum computational strategies to optimize outcomes in the current quantum computing landscape.
>
>
>  **Regarding the large-scaled numerical results:**
>
> The VQLS has demonstrated notable scalability on NISQ devices, particularly when addressing larger problem sizes using artificial examples, as detailed in [4]. This achievement represents a significant stride in establishing the practicality of VQLS-based QSVM on real-world datasets.   This paper presents numerical results and findings as a crucial step towards establishing the practical viability of VQLS-based QSVM.
>
> Our future research endeavors will focus on exploring the scalability of large-scale qubits. This includes identifying and addressing potential challenges while providing analytical insights. Currently, our review of existing literature is centered on the difficulties associated with variational algorithms, such as the barren plateau phenomenon, especially when dealing with a substantial number of qubits.
>
> Undertaking a comprehensive investigation, overcoming these challenges, and analyzing the results for large-scale scenarios will demand additional time and effort. I sincerely hope for your understanding and thanks in advance for your suggestion  in this extended research endeavor.
>
> **Regarding  the limition of condition number:**
>
> The condition number plays a pivotal role in solving linear equations on both classical and quantum computers, as outlined in [5]. For the experimental implementation on HHL-based QSVM [6], the kernel matrix must meet a specific requirement: its condition number needs to be scaled to $poly\log(N)$, where $N$ denotes the size of the linear system [7]. The authors in [4] numerically analyze the time consumption of the VQLS concerning the condition number. In our paper, we highlight a key finding: a small condition number in matrix inversion is able to enhance the trainability of quantum ansatz.
>
> I hope my response has addressed the concerns you raised. I look forward to receiving your feedback.
>
> ---
>
> [1] Lennart Bittel, and Martin Kliesch. Training Variational Quantum Algorithms Is NP-Hard. Physical review letters 2021.
>
> [2] P. Rebentrost, M. Mohseni, and S. Lloyd, “Quantum support vector machine for big data classification,” Physical review letters 2014.
>
> [3] M. Cerezo _et al._, “Variational quantum algorithms,” _Nat Rev Phys_ 2021.
>
> [4] Bravo-Prieto, Carlos, et al. "Variational quantum linear solver." arXiv preprint, 2019.
>
> [5] A. W. Harrow et al., ‘Quantum Algorithm for Linear Systems of Equations’, _Phys. Rev. Lett._, 2009.
>
> [6] Zhaokai Li, et al. " Experimental realization of a quantum support vector machine". Physical review letters, 2015.
>
> [7] Liu, Xiaonan, et al. "Survey on the Improvement and Application of HHL Algorithm." _Journal of Physics, 2022.

---

### Meta-Review · Area_Chair_HkLc · 2023-12-04

**Metareview:**

This paper applies a variational quantum linear system solver to proposing a new quantum algorithm for support vector machines (QSVM). The main strength is having numerical experiments for a larger problem (8-dimensional feature space) from the Iris dataset compared to prior QSVM papers. However, the paper also has weaknesses as it does not provide analytical studies (the result is purely numerical), the idea of combining HHL and QSVM is relatively standard, and the comparison to other quantum machine learning methods is inadequate.

The paper also has a non-anonymity issue by having an acknowledgement section at the top of Page 10 with detailed persons and funding information to acknowledge.

**Justification For Why Not Higher Score:**

The reviewers have common opinions that the paper has several weaknesses: the techniques are relatively standard with novelty not significant, the results are numerical without analytical contributions, and comparisons to other quantum machine learning methods are inadequate. These points together imply that the paper has distance to the standard of ICLR 2024.

For future versions of this paper, the authors may consider further improvements from these perspectives. The review reports also highlight other minor points for the authors to take into consideration in future revisions.

**Justification For Why Not Lower Score:**

N/A

---

### Decision · Program_Chairs · 2024-01-16

Reject